

# Relationship between catheter related cerebrospinal fluid infections and systemic immune-inflammation index

Sebnem Nergiz[1] and Pinar Aydin Ozturk[2]

[1] Department of Dietetics and Nutrition, Dicle University, Ataturk Faculty of Health Sciences, Diyarbakir, Turkey
[2] Department of Neurosurgery, Health Sciences University of Turkey, Diyarbakır Gazi Yasargil Education and Research Hospital, Diyarbakir, Turkey

## ABSTRACT

**Background**. This study investigated the relationship between the systemic immune inflammation index (SII) and catheter-related infections and their effects on prognosis in pediatric patients.

**Methods**. A total of 56 pediatric patients diagnosed with ventriculoperitoneal (V-P) shunt infection between January 2017 and October 2019 were included. V-P shunt infection diagnosis was made based on clinical findings. All cerebrospinal fluid (CSF) samples were subjected to direct microscopic examination and culture. Protein, glucose, and sodium levels in CSF, CSF leukocytes, and hematological and biochemical parameters were measured.

**Results**. Fifty-six patients with growth in CSF culture were included in this study. 55.4% of the cases were female and 44.6% male. V-P shunt was detected in 82.1% of the cases and external ventricular drainage (EVD) catheter-related infection in 17.9%. The CSF/blood glucose ratio was significantly lower ($p = 0.046$), and SII was significantly increased ($p = 0.002$) in non-coagulase-negative staphylococci.

**Conclusions**. Early and appropriate antibiotic therapy reduces morbidity and mortality in catheter-related infections. However, it is important to start empirical antibiotherapy until culture results are expected. Therefore, further research on the estimation of possible factors is needed.

# INTRODUCTION

The central nervous system is protected by the blood–brain barrier against microbial invasion. However, the disruption of the blood–brain barrier could lead to infection (*Couffin et al., 2018*). Ventriculoperitoneal (V-P) shunting is the main treatment method for hydrocephalus. Despite the positive developments in the treatment of hydrocephalus with V-P shunt, complications, especially infections, are the most common problem during treatment. The secondary infection rate due to V-P shunt is 0.3%–40%. Most V-P shunt infections occur within the first month after V-P shunt operations (*Simon et al., 2016*). Young age, prematurity, previous V-P shunt infection, reason for shunt insertion (purulent

Corresponding author
Sebnem Nergiz,
sebnemnergiz@gmail.com

meningitis, intracranial hemorrhage, meningomyelocele), surgeon's experience, duration and technique of surgery, compliance with sterility and three or more shunt changes, the presence of various shunt systems, postoperative cerebrospinal fluid (CSF) leakage, handling of the shunt device, glove holes, poor skin condition of the patient, contamination of the shunt catheter, and excessive personnel entry and exit during the operation are risk factors for the development of V-P shunt infection (*Reddy, Bollam & Caldito, 2012*).

The most common causes of V-P shunt infections are pathogens originating from the skin and enteric system. V-P shunt infections are caused by various pathogens, such as *Staphylococcus aureus*, coagulase-negative staphylococci (CoNS), gram-negative bacteria, and anaerobic bacteria. Among these pathogens, anaerobic and gram-negative bacteria are generally known to cause benign and slow infections. In contrast, gram-positive bacteria cause more aggressive and rapid infections. Among the gram-positive bacteria, *S. aureus* is thought to cause more rapid-onset infections than CoNS (*Tuan et al., 2011*).

External ventricular drainage (EVD) is a system used for intracranial pressure monitoring, CSF drainage in the occluded ventricular system, or the treatment of infected catheters. It is also an important cause of nosocomial meningitis. EVD-related infections have been known to have a rate of 9%–20% (*Dorresteijn et al., 2020*). CoNS, *S. aureus*, and gram-negative bacteria, most commonly *E. coli*, *Acinetobacter* spp., and *Klebsiella* spp., are considered potential bacterial pathogens (*Dorresteijn et al., 2019*).

Platelet activation and leukocytosis occur as a result of the stimulation of the systemic immune reactions after catheter-related central nervous system infections. Increased inflammatory biomarkers have important effects on the severity of the disease, response to treatment, and prognosis in catheter-related central nervous system infections (*Chen et al., 2021*). Recently, several new white blood cells (WBC)-based inflammatory indices namely, neutrophil-to-lymphocyte ratio (NLR), platelet-to-lymphocyte ratio (PLR), have been introduced as prognostic markers. NLR and PLR represent the peripheral neutrophil and lymphocyte counts and the peripheral platelet and lymphocyte counts (*Carpio-Orantes, García-Méndez & Hernández-Hernández, 2020*). The systemic immune-inflammation index (SII) is a new parameter reflecting the balance between inflammation and immune status. SII integrates the lymphocyte, neutrophil, and platelet counts into an index. SII is calculated using the following formula: SII = Platelet × Neutrophil/Lymphocyte (*Chen et al., 2017*).

In recent years, SII has been shown to have a prognostic value in estimating different malignant diseases (*Chen et al., 2017*). However, the prognostic value of SII in central nervous system infections is unknown. The aim of this study is to investigate the relationship between SII and catheter-related central nervous system infections and their effects on prognosis in the pediatric group. This text was previously published as a preprint (*Nergiz & Aydin Ozturk, 2022*).

## MATERIALS & METHODS

Pediatric patients with a diagnosis of V-P shunt or EVD catheter-related central nervous system infection confirmed by growth in CSF cultures between January 2017 and October

2019 were retrospectively included in this study. Patients' age, gender, complete blood count at hospitalization, biochemistry, C-reactive protein (CRP) levels, SII, catheter type, clinical findings, laboratory characteristics, CSF findings (CSF cell count, CSF glucose, CSF sodium, CSF protein levels), CSF culture results (gram staining, resistance), length of hospital stay, treatments (antibiotics, duration), and the final status of the patients were recorded in the follow-up forms. SII was calculated according to the following formula: SII = Platelet × Neutrophil/Lymphocyte. NLR was calculated according to the following formula: NLR = Neutrophil/Lymphocyte. PLR was calculated according to the following formula: PLR = Platelet/Lymphocyte. Patients with positive CSF culture and unavailable data were not included in this study.

The diagnosis of catheter-related central nervous system infection was made based on the results of clinical findings and the CSF examination. And the diagnosis was confirmed by bacterial growth in the CSF culture. No antibiotic therapy was used to prevent infection. The same shunt systems were implanted in all study patients. A pediatric, ultrasmall, flat-bottomed, medium-pressure ventriculoperitoneal shunt systems were used (Desu standard polysulfone ventriculoperitoneal shunt systems; Desu Medical, Ankara, Turkey). Empirical antibiotic therapy was started until the CSF culture results were obtained. Ceftriaxone was started for patients older than 1 month, and a combination of ampicillin and gentamicin was administered to newborns. After the CSF culture results were obtained (usually 3 days later), specific antibiotic therapy was started. CSF culture results were repeated after the patients were treated for an average of 2 weeks. Antibiotics were discontinued, continued, or changed according to the results of the second culture. It was thought that the patient was cleared of infection, with negative CSF culture results after antibiotic therapy and a decrease in white blood cell count in CSF.

## Microbiological and biochemical examination of CSF

All CSF samples taken under sterile conditions and sent to the laboratory were directly subjected to microscopic examination (gram staining) and culture. The detection of the microorganism grown in culture using gram analysis greatly reduces the possibility of contamination. The CSF samples were first centrifuged. The upper fluid was used for biochemical examination, while the bottom sediment was used for microbiological examination. Gram staining was done on a part of the precipitate and examined microscopically. The rest of the colony was cultivated on chocolate agar, blood agar, and eosin methylene blue agar. The plates were incubated at 37 °C for 18–48 h, and the chocolate medium was incubated in an oven with 5% $CO_2$ for 72 h. At the end of the incubation period, the VITEK 2 automated system (bioMérieux, Marcy-l'Étoile, France) was used to identify the colonies with growth at the species level and to determine their antibiotic susceptibility. Antimicrobial susceptibility results were evaluated according to the current Clinical and Laboratory Standards Institute criteria (*Clinical and Laboratory Standards Institute, 2018*). Our study is retrospective. Therefore, consent form was not obtained from the patients. The study was approved by the ethics committee of Health Sciences University of Turkey, Diyarbakir Gazi Yaşargil Education and Research Hospital (132-2022).
**Table 1  Clinical characteristics of patients with infection caused by gram-negative and gram-positive bacteria.**

| Variables | Total (n = 56) | Gram-negative (n = 18) | Gram-positive (n = 38) | p value |
|---|---|---|---|---|
| Age (month) | 24.32 ± 55.79 | 29.11 ± 51.77 | 11.64 ± 36.89 | 0.663 |
| CSF Leukocyte (Count/L) | 1970.45 ± 5993.39 | 1957.69 ± 4603.25 | 162.66 ± 235.43 | 0.995 |
| Protein (mg/dL) | 3887.96 ± 5296.28 | 3890.20 ± 4377,65 | 3887.00 ± 5719.27 | 0.999 |
| Glucose (mg/dL) | 28.36 ± 21.79 | 20.25 ± 16.79 | 32.09 ± 23.07 | 0.121 |
| CSF glucose/blood glucose | 0.31 ± 0.25 | 0.26 ± 0.27 | 0.33 ± 0.25 | 0.419 |
| CSF Sodium (mEq/L) | 137.67 ± 4.63 | 137.50 ± 0.71 | 137.70 ± 5.12 | 0.959 |
| CSF Chloride (mEq/L) | 111.21 ± 7.97 | 115.50 ± 6.61 | 110.07 ± 8.10 | 0.236 |
| WBC ($10^9$/L) | 16.37 ± 8.07 | 17.56 ± 7.27 | 15.80 ± 8.46 | 0.453 |
| CRP (mg/L) | 51.63 ± 65.25 | 62.20 ± 83.62 | 46.82 ± 55.84 | 0.455 |
| Neutrophil ($10^9$/L) | 9.45 ± 7.21 | 9.72 ± 5.67 | 9.32 ± 7.91 | 0.847 |
| Lymphocyte ($10^9$/L) | 5.23 ± 2.95 | 5.499 ± 3.17 | 5.12 ± 2.87 | 0.671 |
| Eosinophil ($10^9$/L) | 0.51 ± 0.72 | 0.24 ± 0.46 | 0.63 ± 0.79 | 0.335 |
| Platelet ($10^9$/L) | 454.86 ± 214.76 | 417.06 ± 210.15 | 472.76 ± 217.35 | 0.369 |
| SII | 1211.01 ± 1493.01 | 937.99 ± 747.74 | 1333.14 ± 1721.02 | 0.369 |
| NLR | 2.91 ± 3.96 | 2.71 ± 3.20 | 3.00 ± 4.3 | 0.782 |
| PLR | 114.20 ± 80.98 | 96.10 ± 55.32 | 122.30 ± 99.60 | 0.192 |

Notes.

CSF, cerebrospinal fluid; WBC, white blood cell; CRP, C-reactive protein; SII, systemic immune-inflammation index; NLR, neutrophil-to-lymphocyte ratio; PLR, platelet-to-lymphocyte ratio.

## Statistical analyses

Statistical analysis was performed using the SPSS statistical program (Version 12.0; SPSS Inc., Chicago, IL, USA). All clinical parameters were analyzed. The Kolmogorov–Smirnov test was used to determine the normal distribution of the data. The continuous variables were expressed as mean ± SD. The one-way analysis of variance was used to compare the mean of the normally distributed parameters; otherwise, the nonparametric Kruskal–Wallis H-test was utilized. The categorical parameters were expressed as percentages and absolute frequencies. Chi-square test was utilized to compare this type of data. Spearman Rank correlation analysis was performed. A p value of <0.05 was considered statistically significant.

## RESULTS

Fifty-six patients with growth in CSF culture were included in this study. 55.4% of the cases were female and 44.6% male. The youngest of the cases was 1, and the oldest was 216 months old. The mean age was 24.32 (± 55.79). Ventriculoperitoneal shunt was detected in 82.1% of the cases, and EVD catheter-related infection was found in 17.9%. Gram-positive bacteria were grown in 67.9% of the cases and gram-negative bacteria in 32.1%. When the infection and CSF biochemistry parameters of gram-positive and gram-negative bacteria were compared, no significant difference was observed (Table 1).

The prevalence rates of the bacteria are as follows: 41.1% of CoNS; 10.7% of *Klebsiella pneumoniae* and *Enterococci*; 7.1% of *S. aureus*; 5.4% of *Acinetobacter baumannii*,

**Table 2** Clinical characteristics of patients with infection caused by gram-negative bacteria, coagulase negative staphylococci (CoNS) and non-CoNS gram-positive bacteria.

| Variables | Gram-negative ($n = 18$) | Non-CoNS gram-positive ($n = 15$) | CoNS ($n = 23$) | p value |
|---|---|---|---|---|
| Age (month) | 29.11 ± 51.77 | 25.67 ± 51.68 | 19.70 ± 52.85 | 0.865 |
| Cell | 1957.69 ± 4603.25 | 694.48 ± 821.47 | 2871.30 ± 8568.33 | 0.779 |
| CSF Protein (mg/dL) | 3890.20 ± 4377.65 | 1836.14 ± 2520.53 | 4570.62 ± 6347.43 | 0.509 |
| CSF Glucose (mg/dL) | 20.25 ± 16.79 | 14.42 ± 15.38 | 38.61 ± 22.23 | 0.009 |
| CSF glucose/blood glucose | 0.26 ± 0.27 | 0.14 ± 0.14 | 0.40 ± 0.24 | 0.046 |
| CSF Sodium (mEq/L) | 137.50 ± 0.71 | 139.00 ± 4.24 | 137.38 ± 5.52 | 0.921 |
| CSF Chloride (mEq/L) | 115.50 ± 6.61 | 108.50 ± 12.08 | 111.11 ± 4.54 | 0.419 |
| WBC ($10^9$/L) | 17.56 ± 7.27 | 17.86 ± 11.33 | 14.47 ± 5.83 | 0.343 |
| CRP (mg/L) | 62.20 ± 83.62 | 55.23 ± 59.13 | 41.35 ± 54.44 | 0.638 |
| Neutrophil ($10^9$/L) | 9.72 ± 5.67 | 12.37 ± 10.94 | 7.34 ± 4.31 | 0.107 |
| Lymphocyte ($10^9$/L) | 5.499 ± 3.17 | 4.15 ± 3.09 | 5.74 ± 2.59 | 0.244 |
| Eosinophil ($10^9$/L) | 0.24 ± 0.46 | 0.35 ± 0.53 | 0.51 ± 0.91 | 0.505 |
| Platelet ($10^9$/L) | 417.06 ± 210.15 | 467.93 ± 269.06 | 475.91 ± 182.65 | 0.667 |
| SII | 937.99 ± 747.74 | 2302.14 ± 2374.27 | 701.19 ± 580.02 | 0.002 |
| NLR | 2.71 ± 3.20 | 5.44 ± 6.08 | 1.40 ± 0.92 | 0.007 |
| PLR | 96.10 ± 55.32 | 168.20 ± 119.53 | 92.36 ± 45.14 | 0.008 |

Notes.
CSF, cerebrospinal fluid; WBC, white blood cell; CRP, C-reactive protein; SII, systemic immune-inflammation index; NLR, neutrophil-to-lymphocyte ratio; PLR, platelet-to-lymphocyte ratio.

*Pseudomonas aeruginosa*, and *Serratia marcescens*; and 1.8% of *Bacillus* spp., *Brevundimonas diminuta*, *Escherichia diminutoantea*, *Kocrychia cocuricaelus*, Koccurichia cocurica, *Streptococcus salivarius*, and *Granuli elegans*.

When gram-negative and gram-positive bacteria were compared according to coagulase status, it was found that the CSF/blood glucose ratio was statistically significantly lower ($p = 0.046$) and SII was significantly higher in the non-CoNS gram-positive bacteria ($p = 0.002$). Also, NLR and PLR were significantly higher in the non-CoNS gram-positive bacteria ($p = 0.007$ and $p = 0.008$) (Table 2).

CoNS bacteria have a higher resistance to methicillin, erythromycin, and clindamycin than the other groups ($p = 0.001$, $p = 0.006$, $p = 0.044$, respectively). Gram-negative bacteria have a high resistance to ampicillin, 3rd and 4th generation cephalosporin, broad-spectrum beta-lactamase ($p = 0.041$, $p = 0.000$, $p = 0.028$, $p = 0.028$, respectively) (Table 3).

Single antibiotics were started in 32.1% of the patients, double antibiotics in 66.1%, and triple antibiotics in 1.8% (relationship between the groups, $p = 0.571$). Antibiotic revision was required in 16.1% of the patients. Revision was required once in 7.1%, twice in 7.1%, and thrice in 1.8%. Only one patient (1.8%) required intrathecal treatment. The mortality rate was 19.6%. Excluding the patients who did not respond to treatment (those that resulted in mortality), the mean duration of treatment was 20.42 ± 11.45 days. The mean duration of treatment was 21.69 ± 9.44 days in the gram-negative bacteria group,

**Table 3** CSF, cerebrospinal fluid; WBC, white blood cell; CRP, C-reactive protein; SII, systemic immune-inflammation index; NLR, neutrophil-to-lymphocyte ratio; PLR, platelet-to-lymphocyte ratio.

| Resistance | Gr (-) (%)[a] | Non-CoNS Gr (+) (%)[a] | CoNS (%)[a] | Total (%) | p value |
|---|---|---|---|---|---|
| Methicillin | 0 | 26.7 | 52.2 | 28.5 | 0.001 |
| Vancomycin | 0 | 13.3 | 0 | 3.6 | 0.065 |
| Erythromycin | 0 | 6.7 | 30.4 | 14.3 | 0.006 |
| Aminoglycoside | 17.6 | 6.7 | 8.7 | 10.7 | 0.613 |
| Clindamycin | 5.5 | 13.3 | 53.3 | 19.6 | 0.044 |
| Ampicillin | 22.2 | 0 | 4.3 | 8.9 | 0.041 |
| 3rd generation cephalosporin | 38.9 | 0 | 0 | 12.5 | 0.000 |
| 4th generation cephalosporin | 16.7 | 0 | 0 | 5.4 | 0.028 |
| Trimethoprim | 11.1 | 0 | 0 | 3.6 | 0.095 |
| ESBL | 16.7 | 0 | 0 | 5.4 | 0.028 |
| Colistin | 5.6 | 0 | 0 | 1.8 | 0.315 |
| Gentamicin | 11.1 | 6.7 | 13.0 | 10.7 | 0.811 |
| Penicillin | 5.6 | 20 | 4.3 | 8.9 | 0.255 |
| Tazobactam | 5.6 | 0 | 0 | 1.8 | 0.315 |
| Quinolone | 27.8 | 6.7 | 30.4 | 23.2 | 0.149 |
| Tetracycline | 0 | 0 | 4.3 | 1.8 | 0.405 |

**Notes.**
[a] It was calculated as the subgroup ratio.

**Table 4** The SII values of gram-negative bacteria, coagulase-negative staphylococci (CoNS), and non-CoNS gram-positive bacteria according to mortality.

| Bacteria group | SII in treatment | SII in mortality | p value |
|---|---|---|---|
| Gram-negative | $638.51 \pm 553.49$ | $1656.74 \pm 692.25$ | 0.006 |
| Non-CoNS gram-positive | $2382.14 \pm 2481.61$ | $1782.14 \pm 2096.49$ | 0.753 |
| CoNS | $548.42 \pm 373.43$ | $1426.82 \pm 883.47$ | 0.003 |

**Notes.**
SII, systemic immune-inflammation index; CoNS, coagulase-negative staphylococci.

$23.77 \pm 12.57$ days in the gram-positive and non-CoNS group, and $17.26 \pm 11.67$ days in the gram-positive and coagulase-negative group, indicating no statistically significant difference ($p = 0.263$). No statistical difference was found in all groups regarding the need for multiple antibiotics, the need for antibiotic revision, mortality, and response time to treatment.

Considering the correlation between SII and mortality between the groups; in the CoNS group and the gram-negative bacteria group, the SII was found to be statistically significantly higher in those that resulted in mortality. However, in the non-CoNS gram-positive bacteria group, no significant difference in SII was found between the patients who survived and the patients who died (Table 4). In accordance with the ROC curve of SII to estimate mortality, when 933.35 was set as the cut-off point, SII demonstrated the best estimating mortality with the ROC analysis (sensitivity: 72.7% and specificity: 70.5%, AUC: 0.744).

**Table 5   Correlation analysis between SII and clinical parameters in patients with V-P shunt infections.**

| Parameters | Sperman's correlation coefficient ($r$) | $p$ value |
|---|---|---|
| CSF Cells | 0.415 | 0.049 |
| CSF Glucose | −0.326 | 0.035 |
| CSF Sodium | −0.621 | 0.042 |
| CSF Chloride | −0.500 | 0.034 |
| WBC | 0.547 | <0.001 |
| Neutrophil | 0.820 | <0.001 |
| Lymphocyte | −0.331 | 0.014 |
| Eosinophil | −0.273 | 0.044 |
| Platelet | 0.357 | 0.007 |
| Antibiotic revision | 0.672 | 0.033 |
| Mortality | 0.338 | 0.012 |

Notes.

SII, systemic immune-inflammation index; CSF, cerebrospinal fluid; WBC, white blood cell.

SII positively strongly correlated with neutrophil and antibiotic revision. SII positively weak correlated with platelet and mortality. SII positively moderate correlated with CSF cells and WBC. However, SII negatively weak correlated with CSF glucose, CSF sodium, CSF chloride, lymphocyte and eosinophil. SII negatively moderate correlated with CSF sodium and CSF chloride (Table 5).

## DISCUSSION

Central nervous system infections are a serious group of infections that cause morbidity and mortality, and this can be prevented with early and appropriate antibiotic therapy. Empirical antibiotic treatment were started according to the results of the CSF microbiological examination, infection parameters, and culture results (*Lu et al., 2002*). Serum inflammatory markers that make up the SII are parameters that can be easily obtained in clinical practice. SII has a prognostic value in various studies (*Chen et al., 2021*). However, no study has investigated the relationship between SII and bacterial agents in catheter-related infections. In the present study, the relationship between SII and possible bacterial agents was evaluated, aiming to provide appropriate antibiotic therapy by determining the parameters at the time of application and possible factors before the culture results were obtained. At the same time, the prognostic effect of SII among bacterial groups was evaluated.

Catheter-related central nervous system infections, especially V-P shunt infections, are more common in the pediatric age group (*Kestle et al., 2011*). CSF biochemical examination together with clinical findings can be useful in the diagnosis of central nervous system infections. Moreover, a central nervous system infection is characterized by a white blood cell count of >10/mm3, decreased glucose levels (<45 mg/dL), and high protein levels (>100 mg/dL) in CSF (*Yakut et al., 2018*). In the present study, CSF WBC was determined as 1970.45 ±5993.39/mm$^3$ (50-27249), CSF glucose 28.36 ± 21.79 mg/dL (1-91), and

CSF protein 3887.96 ± 5296.28 mg/dL (4-20723). The lowest CSF glucose level and simultaneous CSF/blood glucose ratio were observed in the non-CoNS gram-positive bacteria group, whereas the highest CSF glucose level and simultaneous CSF/blood glucose ratio were observed in the CoNS group, indicating a statistically significant difference ($p = 0.009$–$0.046$.)

Shunt infections are associated with a high revision rate, recurrent infections, ventriculitis, meningitis, encephalitis, and increased mortality (*Winston, Ho & Dolan, 2013*). They include shunt removal, EVD insertion, and antibiotic therapy (*Rehman et al., 2010*). Complications include decreased intelligence quotient and increased risk of seizures due to neuronal damage and infection (*Vinchon & Dhellemmes, 2006*). The most common causative agent in shunt infections is CoNS, which is found in the skin flora (*Schreffler, Schreffler & Wittler, 2002*). CoNS constitutes 50% and *S. aureus* 25% of the bacterial agents (*Stevens et al., 2012*). Gram-negative bacteria are less common bacteria, whereas *Pseudomonas* spp., *Klebsiella* spp., and *Acinetobacter* spp. are common bacteria (*Yakut et al., 2018*). The risk factors for the development of EVD-associated infection are blood mixing in CSF, CSF sampling frequency, CSF leakage at the drainage entry site, and possibly bilateral drainage (*Van de Beek, Drake & Tunkel, 2010*). Younger age (<1 year and prematurity), decreased skin integrity, and intraventricular hemorrhage in children increase the risk of EVD-associated infection compared with adults (*Dorresteijn et al., 2020*; *Zheng et al., 2018*). The most common factors of EVD catheter-related infection are CoNS, *S. aureus*, and gram-negative bacteria, *i.e., E. coli*, *Acinetobacter* spp., and *Klebsiella* spp. (*Dorresteijn et al., 2020*). In the present study, CoNS was the most common, followed by *Klebsiella pneumoniae*, *Enterococci*, *S. aureus*, *Acinetobacter baumannii*, *Pseudomonas aeruginosa*, and *Serratia marcescens*, respectively. The rate of gram-positive bacteria was 67.9%, whereas the rate of gram-negative bacteria was 32.1%.

SII is widely used in various tumors as a prognostic indicator (*Gu et al., 2020*). An increased level of SII is associated with a poor prognosis (*Trifan & Testai, 2020*). In the present study, the effect of SII on prognosis in the bacterial group was examined. It was found that the SII was not predictive in distinguishing between gram-negative and -positive bacteria.

CoNS is a group of bacteria found in the skin mucosa, which is considered apathogenic in healthy individuals. However, it is a common cause of hospital infections and catheter-related infections. It usually presents as subacute and chronic infections that start subclinically. However, it can have an aggressive course and be fatal in those who are inadequately treated. It should be considered in the foreground of foreign body-related infections (*Heilmann, Ziebuhr & Becker, 2019*). It was determined that the SII, NLR and PLR were significantly increased in the gram-positive bacteria group without CoNS ($p < 0.05$). However, in the non-CoNS gram-positive bacteria group, an inverse correlation was found between mortality and SII, which was not statistically significant. However, it was observed that the increased SII was significantly associated with mortality in the CoNS and gram-negative bacteria groups ($p < 0.05$). Gram-negative bacteria and CoNS usually cause benign and slow infections. The fact that the SII was higher in gram-negative bacteria and CoNS patients with mortality than in survivors may be due to the more severe and

progressive gram-negative infections in the mortality group. Moreover, the reason SII was not different between survivors and patients with mortality in the non-CoNS gram-positive bacteria group and the SII was higher compared with the other groups may be due to the fact that these infections were more aggressive and rapid.

Antibiotic resistance changes the treatment response and modalities. While methicillin resistance is prominent in both CoNS and *S. aureus*, broad-spectrum beta-lactamase resistance and carbapenem resistance cause problems in gram-negative bacteria (*Chaurasia, Shinde & Baveja, 2021*). In the present study, the rate of resistance to any antibiotic was found to be 72.3%, with methicillin resistance being the highest (28.5%). While methicillin resistance was 52.2% in CoNS, it was 26.7% in non-CoNS gram-positive bacteria. Although the rate of methicillin resistance has been increasing (*Lee et al., 2012*), it was detected at a lower rate in the present study. ESBL resistance was found to be 16.7% in gram-negative bacteria. Clindamycin and quinolone have been found to be highly resistant.

The limitation of the our study is the small number of samples analyzed.

## CONCLUSIONS

Early and appropriate antibiotic therapy reduces morbidity and mortality in catheter-related infections. However, it is essential to start empirical antibiotic therapy until the culture results are determined. Therefore, further research on the estimation of possible factors is important. In the present study, it was determined that the level of SII upon admission was mostly parallel with non-CoNS gram-positive bacteria. However, high SII upon presentation was associated with poor prognosis and increased mortality in both the CoNS and gram-negative bacteria groups. Furthermore, no bacterial difference was observed in the shunt or EVD-related infections.

### Funding
The authors received no funding for this work.

### Competing Interests
The authors declare there are no competing interests.

### Author Contributions
- Sebnem Nergiz conceived and designed the experiments, performed the experiments, analyzed the data, prepared figures and/or tables, authored or reviewed drafts of the article, and approved the final draft.
- Pinar Aydin Ozturk conceived and designed the experiments, performed the experiments, analyzed the data, prepared figures and/or tables, authored or reviewed drafts of the article, and approved the final draft.

### Human Ethics
The following information was supplied relating to ethical approvals (*i.e.*, approving body and any reference numbers):

Health Sciences University of Turkey, Diyarbakir Gazi Yaşargil Education and Research Hospital (Ethical Application Ref:132-2022).

## Ethics

The following information was supplied relating to ethical approvals (*i.e.*, approving body and any reference numbers):

Health Sciences University of Turkey, Diyarbakir Gazi Yaşargil Education and Research Hospital (Ethical Application Ref:132-2022).

## Data Availability

The raw data is available in the Supplemental File.

## Supplemental Information

Supplemental information for this article can be found online at http://dx.doi.org/10.7717/peerj.15905#supplemental-information.

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
