# Peer review of "Relationship between catheter related cerebrospinal fluid infections and systemic immune-inflammation index"

_PeerJ, doi:10.7717/peerj.15905_

## Round 0.1 · original submission · Minor Revisions

Please address the reviewer's comments for minor revisions.

·

Basic reporting

The manuscript is clearly written with adequate background and references.

Experimental design

The authors identify different bacterial infections from the CSF of patients and attempts to find its correlation to established measures of immune inflammation, I.e., NLR, PLR, SII, etc.

The authors claim through adequate background that the correlation between these parameters are studied in the context of tumors, but the same in infection is lacking.

Finding this correlation will help treat patients facing CSF infections before the bacterial culture results are made available.

Minor concerns:
the study is reporting measures of multiple different parameters, but the reasoning for choosing said parameter are not well justified. I recommend removing parameters in the “variables” column of table 1 and 2 that show non-significant p-value.

Validity of the findings

The authors show that the patients with gram negative bacteria and the CoNS gram positive bacterial infections show lower SII than non-Co-NS infected patients (Table 2).

They also show reduced SII during treatment in comparison to patients with mortality for gram-negative and Co-NS infected patients (Table 4)

Minor revision:
A) Hu, B., et. Al, 2014 clinical cancer research paper that defined SII in HCC used x-tile analysis to determine that SII>300 x 10(9) is poor prognosis for HCC patients. I recommend the authors to try a similar approach for each bacterial group to determine the SII value that could mean increased CSF infection.

B) Although there are both positive and negative correlations reported between SII and clinical parameters in table 5, the strongest correlation is observed only in neutrophil level and antibiotic revision. While the weak correlation of SII were with CSF glucose, lymphocyte, eosinophil, platelet and mortality. Rest of the parameters in Table 5 show moderate correlation. I recommend the authors to correct the last paragraph of the results section accordingly, as in the current form it is very misleading.

Additional comments

Interesting work with potential.

Reviewer 2 ·

Basic reporting

The authors have provided adequate literature references. The structure of the paper is defined.

Experimental design

The limitation of the paper is the number of samples analyzed. Please include it as a limitation of the study in discussion section.

Validity of the findings

The validity of the findings is not assessed.

---

## Round 0.2 · accepted · Accept

I request that the authors update the abstract conclusions in the final proof to reflect what the reviewer has requested. And, congratulations on your article's acceptance!

·

Basic reporting

Minor revision:

The “Conclusion” in the Abstract is not accurate and in its current form is basically just “future direction. I reccommend the authors to paraphrase it to something similar to lines 261-265.

Experimental design

No comments

Validity of the findings

No comments

Additional comments

No comments

Reviewer 2 ·

Basic reporting

The authors have included statements in the manuscript according to the reviewers' suggestion.

Experimental design

No comment

Validity of the findings

No comment